# Genetic Diversity of Bovine Group A Rotavirus Strains Circulating in Korean Calves during 2014 and 2018

**DOI:** 10.3390/ani12243555

**Published:** 2022-12-15

**Authors:** Gyu-Nam Park, SeEun Choe, Ra Mi Cha, Jihye Shin, Ki-Sun Kim, Byung-Hyun An, Song-Yi Kim, Bang-Hun Hyun, Dong-Jun An

**Affiliations:** 1Virus Disease Division, Animal and Plant Quarantine Agency, Gimcheon 39660, Republic of Korea; 2Department of Veterinary Medicine Virology Laboratory, College of Veterinary Medicine and Research Institute for Veterinary Science, Seoul National University, GwanAk-Ro 1, GwanAk-Gu, Seoul 08826, Republic of Korea

**Keywords:** BoRVA, phylogenetic tree, calf, diarrhea, genotype

## Abstract

**Simple Summary:**

Morbidity and mortality rates due to bovine group A rotavirus (BoRVA) infections are high, resulting in direct and indirect economic losses to the beef and dairy industries. RVA strains, which are antigenically heterogeneous, are classified into multiple G and P types; these two types are defined by the two outer capsid proteins VP7 and VP4, respectively. A clear understanding of the various VP7 and VP4 type-specificities is required to ascertain whether it is necessary to construct polyvalent RVA vaccines for calves. In the past, BoRVA G8P[7] was the predominant type in Korean calves; however, we consistently identified G6P[5] as the main cause of diarrhea in young calves from 2014 to the present day.

**Abstract:**

The purpose of this study was to investigate annual changes in BoRVA strains by examining the VP4 and VP7 genes of rotaviruses in Korean calves. Between 2014 and 2018, 35 out of 138 samples of calf diarrhea feces collected nationwide were positive for BoRVA. Further genetic characterization of the VP7 and VP4 genes of 35 BoRVA isolates identified three different G-genotypes (G6, G8, and G10) and two different P genotypes (P[5] and P[11]). The G6 genotype was most common (94.3%) in BoRVA-positive calves, followed by the P[5] genotype (82.9%). Four genotypes comprised combinations of VP4 and VP7: 80% were G6P[5], 14.2% were G6P[11], 2.9% were G8P[5], and 2.9% were G10P[11]. Susceptibility to infection was highest in calves aged < 10 days (35%) and lowest in calves aged 30–50 days (15.4%). The data presented herein suggest that the G6P[5] genotype is the main causative agent of diarrhea in Korean calves. In addition, it is predicted that G6P[5] will continue to act as a major cause of diarrhea in Korean calves.

## 1. Introduction

Rotavirus A (RVA) is the leading cause of acute gastroenteritis in children and young animals worldwide. Bovine group A rotavirus (BoRVA) is one of the main global etiological agents of neonatal diarrhea in calves. Morbidity and mortality rates are high, resulting in direct and indirect economic losses to the beef and dairy industries [1].

Rotavirus (RV), which is a member of the family Reoviridae, comprises a dsRNA genome surrounded by a triple-layered protein capsid. The segmented dsRNA genome encodes six structural proteins (VP1–VP4, VP6, and VP7) and five/six nonstructural proteins (NSP1–NSP5/6) [2,3]. RVs are classified into 10 distinct groups/species (A–J) [3,4]. All have two outer capsid proteins, VP7 and VP4, which are antigenic and induce neutralizing antibodies upon infection; these proteins define the G and P genotypes, respectively [2]. VP4, which is located on the surface of mature virus particles, independently elicits neutralizing antibodies and induces protective immunity [2]. VP7 is the outer capsid protein and participates in both a membrane-displacing assembly step and a membrane-disrupting entry step [2].

Thus far, at least two genotypes, 42G and 58P, have been described in humans and animals worldwide (https://rega.kuleuven.be/cev/viralmetagenomics/virus-classification/rcwg (accessed on 12 October 2022).

Although not much research has been done to date, bovine coronaviruses, bovine viral diarrhea viruses, and RVs have been identified as important diarrhea-causing viruses prevalent in Korean calves [5,6,7,8,9,10]. Among these various diarrhea-causing pathogens, bovine rotavirus (BoRVA) is the primary causative agent in calves aged less than one month [5,11].

G6P[5] is a common genotype of bovine RVA strain found in cattle herds from Asia [12], Europe [13], Oceania [14], and America [15,16]. In South Korea, the 2004–2005 survey did not identify G6P5 [9], but in the 2019–2020 survey, it was isolated at a high rate (58.7% (37/63 samples) [10].

Here, we investigated the appearance time and diversity of the G6P[5] genotype in calf diarrhea fecal samples collected nationwide between 2014 and 2018. We also investigated the age at which calves were most susceptible to infection with BoRVA.

## 2. Materials and Methods

### 2.1. Sample Collection

From 2014 to 2018, 138 calf diarrhea samples from eight regions (excluding Jeju Island) across the country were collected by five large-animal clinical veterinarians: 16 in 2014, 19 in 2015, 41 in 2017, 39 in 2017, and 23 in 2018. According to region, 22 were collected in Gyeonggi, 8 in Gangwon, 23 in Chungnam, 10 in Chungbuk, 29 in Gyeongbuk, 20 in Gyeongnam, 15 in Jeonbuk, and 11 in Jeonnam. Of the 138 calves, 46 were born to cows vaccinated against BoRVA and 92 were born to non-vaccinated cows. The calf age of the collected samples was divided into four groups: 20 from calves aged 1–10 days, 83 from calves aged 11–20 days, 22 from calves aged 21–30 days, and 13 from calves aged 31–50 days. All cattle from which samples were taken showed signs of diarrhea and dehydration, along with anorexia and inactivity. Respiratory symptoms were also observed in some calves.

### 2.2. RNA Extraction and BoRVA Detection

Samples were diluted 1:3 with phosphate-buffered saline (PBS) and centrifuged at 3000× *g* for 20 min. The supernatants were filtered through a 0.22 μm-pore-syringe filter (Minisart^®^ syringe filter, Sartorius, Göttingen, Germany) and stored at −80 °C until further analysis. Total RNA was extracted from each supernatant using an RNeasy mini kit (Qiagen Inc., Germantown, MD, USA) and eluted with 50 μL of elution buffer. The samples were first screened for expression of the VP6 gene using real-time, one-step RT-PCR [17], and the amplification values (Ct values) were determined for each sample. Ct values ≤ 33 indicated a positive result.

### 2.3. Genotyping and Sequencing

Thirty-five positive RNA samples obtained from real-time RT-PCR screening were reverse-transcribed using the HelixCript^TM^ easy cDNA synthesis kit (Nonohelix Co., Ltd., Republic of Korea). The cDNA from BoRVA-positive samples was subjected to G and P genotyping to detect a VP7 gene fragment of 887 bp and a VP4 gene fragment of 664 bp, as described previously [18,19]. PCR assays were carried out using the *AccuPower*^®^ ProFi Taq PCR PreMix kit (Bioneer Inc., Daejeon, Republic of Korea), as described previously [18,19]. The amplified PCR products were extracted using the Omega Gel kit (Omega) and subcloned into the pGEM-T vector system II (Promega, Madison, WI, USA). Cloned plasmids containing the BoRVA VP7 and VP4 genes were sequenced with T7 and SP6 sequencing primers and an ABI Prism 3730xl DNA sequencer (Cosmo Genetech Co., Seoul, Republic of Korea). Next, the Basic Local Alignment Search Tool (BLAST, http://blast.ncbi.nlm.nih.gov/Blast.cgi (accessed on 19 October 2022)) program was used to compare nucleotide sequences with reference sequences available in the GenBank database.

### 2.4. Phlyogenetic Analysis

The genotypes of the two major genome segments (VP7 and VP4) were determined using nucleotide BLAST and a web-based RotaC genotyping tool: http://viprbrc.org (accessed on 18 October 2022) [20]. Multiple sequence alignment (including reference sequences) was performed using CLUSTAL X (ver. 2.1) (http://www.clustal.org/clustal2/, accessed on 18 October 2022). Phylogenetic trees based on the partial nucleotide sequences of VP7 and VP4 were constructed using the maximum-likelihood (ML) method and Molecular Evolutionary Genetics Analysis X (MEGA X) software (https://www.megasoftware.net/, accessed on 18 October 2022) [21], with nucleotide distance (*p*-distance) and 1000 replications used for bootstrap analysis. Before each phylogenetic analysis, the model of nucleotide substitution that best fitted the data was identified using the “find best model” function in MEGA X. Phylogenetic trees were constructed based on the partial VP7 and VP6 gene sequences of BoRVA reference strains isolated in Asia, America, and Europe, including the Korean strains detected in 2004–2020. Partial VP7 and VP4 gene sequences of BoRVA reference strains were obtained from the National Center for Biotechnology Information (NCBI) GenBank database.

## 3. Results

### 3.1. Detection of Bovine Rotavirus Antigens

The BoRVA-positive rate in the 138 calf diarrheic fecal samples collected between 2014 and 2018 was 25.4% (35/138) (Table 1). The yearly BoRVA-positive rates were as follows: 25% (4/16) in 2014, 26.3% (5/19) in 2015, 19.5% (8/41) in 2016, 30.8% (12/39) in 2017, and 20.1% (6/23) in 2018 (Table 1). According to geographical region, the BoRVA-positive rates over the 5 years were as follows: 27.3% (6/22) in Gyeonggi, 12.5% (1/8) in Gangwon, 30.4% (7/23) in Chungnam, 20.0% (2/10) in Chungbuk, 27.6% (8/29) in Gyeongbuk, 25.0% (5/20) in Gyeongnam, 26.7% (4/15) in Jeonbuk, and 18.2% (2/11) in Jeonnam (Table 2 and Table 3). Nationwide, the G6 genotype was predominant; however, G10P[11] was also detected in Gyeonggi and G8P[5] was detected in Jeonbuk (Table 2 and Table 3). The BoRVA-positive rate was the highest (35% (7/20)) between Days 1 and 10 post-birth (Table 2 and Table 3). 

The BoRVA-positive rates after this time were as follows: 25% (20/80) between 11 and 20 days, 27.3% (6/22) between 21 and 30 days, and 15.4% (2/13) between 31 and 50 days (Table 2 and Table 3). Interestingly, G8P[5] and G10P[11] were detected in calves aged 1–10 days (Table 2 and Table 3). The BoRVA-positive rates in calves born to vaccinated and non-vaccinated cows were 26.1% (12/46) and 25.0% (23/92), respectively (data not shown).

### 3.2. G and P Genotypes of Korean BoRVA

To determine the G and P genotypes, the 35 BoRVA-positive fecal samples from calves with suspected rotaviral enteritis were subjected to partial nucleotide sequencing and phylogenetic analysis of the VP7 and VP4 genes (Table 1). The G6, G8, and G10 genotypes were the most prevalent. Among them, G6 was the predominant VP7 genotype, accounting for 94.2% (33/35) of strains (Table 2). The most prevalent VP4 genotypes were P[5] (85.7%, 30/35) and P[11] (14.3%, 5/35) (Table 2). With respect to the prevalence of combined G and P genotypes over 5 years, G6P[5] was the highest (80% (28/35)), followed by G6P[11] (14.2% (5/35)) and G8P[5] and G10P[11] (both 2.9% (1/35)) (Table 1 and Figure 1B). Previous studies report that the prevalence of combined G and P BoRVA genotypes in South Korea in 2004–2005 [9] was as follows: 90% (27/30) for G8P[7] and 10% (3/30) for G6P[7]. 

However, in 2019–2020 [10], this changed to 58.7% (37/63) for G6P[5], 3.2% (2/63) for G6P[11], 6.3% (4/63) for G8P[5], 1.6% (1/63) for G10P[11], 1.6% (1/63) for G5P[1], 27.0 (17/63) for GXP[1], and 1.6% (1/63) for GXP[11] (Figure 1A,C). In addition, the G6P[5] genotype, which is currently the most prevalent strain in South Korea, was first detected in calf diarrhea samples collected from four regions (Gyeonggi, Gyeongbuk, Gyeongnam, and Chungnam) in 2014 (Table 2).

### 3.3. Phylogenetic Analysis of the G Genotype

Phylogenetic analysis of the partial nucleotide sequences of the VP7 gene (an 887 bp fragment) from the 35 Korean BoRVA strains, as well as selected reference strains from GenBank, was performed using the ML method in MEGA X (Figure 2). The phylogenetic tree for the VP7 genes revealed that the 35 Korean BoRVAs belonged to the G6, G8, and G10 types. In particular, 32 of 33 Korean BoRVAs belonging to the G6 type were closely related to USA strains UK, WC, and NCDV; the Chinese strain LNA5; the French strain V019; and the Brazilian strain 91976-SC. 

However, and unusually, one strain (16CN07) was separated slightly, forming a more minor group closer to the USA strain MC27, the Japanese strain KN-4, and the Korean strain KNU-GC14, reported in 2020 [10]. One Korean BoRVA belonging to the G8 type was closely related to the Japanese strains GB20-25 and Sun9 and to the Korean strain KJ45, reported in 2006 [9]. One Korean BoRVA belonging to the G10 type was closely related to the USA strain B223, the Brazilian strain 1979-PR, and the Korean strain KNU-YJ8, reported in 2020 [10]. However, although the G5 genotype was detected in 2004, 2006, 2009, and 2020, it was not detected in 2014–2018 in the present study (Figure 2). The accession numbers of the reference sequences reported worldwide, including the Korean strains, are displayed in Figure 2 and Figure 3.

### 3.4. Phylogenetic Analysis of the P Genotype

Phylogenetic analysis of partial nucleotide sequences of the VP4 gene (a 664 bp fragment) from the 35 Korean BoRVA strains and selected reference strains from GenBank was also performed (Figure 3). The phylogenetic tree for VP4 revealed that the 35 Korean BoRVAs belonged to only two types (P[5] and P[11]) and that the P[5] genotype, which contains 30 of the Korean BoRVAs, was closely related to the USA strains Rota Teq-SC2-9 and WC, the Turkish strain P4 Calf TR 2, the French strain V026, the United Kingdom strain, the South Africa strain 1603, and the Thailand strain 61A (Figure 3). In addition, five Korean BoRVAs belonging to the P[11] genotype were closely related to the Japanese strains GB20-25 and GB14-45, the Iranian strain Khorasan, the Brazilian strain 1929-PR, the Chinese strain DQ-75, and the US strain B223. However, although the P[7] genotype was detected in 2004, 2006, and 2020, it was not detected in 2014–2018 in the present study (Figure 3).

## 4. Discussion

More than 50% of the mortality observed in pre-weaned calves is related to diarrhea, and most cases occur in calves aged less than one month [22]. A previous study of BoRVA-positive Korean native calves reported that 17.1% (28/164) of diarrhea samples were obtained in Gangwon and Gyeongbuk provinces between 2014 and 2016 from calves aged less than 7 months [5]. Another study of fecal samples collected in 2016–2017 from Korean calves aged up to 60 days showed that the BoRVA-positive rate for diarrhea feces was 15.2% (31/204), compared with 5.0% (17/340) for normal feces [6]. In addition, we found a BoRVA-positive rate of 25.4%, with the prevalence being above this average in four regions (Chungnam, Gyeongbuk, Gyeonggi, and Jeonbuk). Another study reported that diarrhea samples collected from calves in Jeonbuk and Gyeongbuk in 2019–2020 showed a BoRVA-positive rate of 42.5% [10]. In the present study, we found that 35.0% (7/20) of BoRVA-positive diarrheal feces were collected from calves aged 1–10 days, which is lower than the 47.2% positive rate reported in the previous study [10]; however, both studies suggest that BoRVA is most prevalent in calves under 10 days of age.

In China, which is geographically close to South Korea, several G genotypes (G6, G8, and G10) and P genotypes (P[1], P[5], P[7], and P[11]) have been detected in the bovine population [23,24,25]. The only relevant study previously conducted in China showed that G6 and P[5] were common G and P genotypes among dairy calves in some regions and that the dominant genotype combination was G6P[5] [23,25]. Similar to the G and P genotypes of BoRVA occurring in China, we found that genotypes G6, G8, G10, P[5], and P[5] were prevalent in Republic of Korea from 2014 to the present day and that G5 emerged in 2020 [10]. However, the P[7] genotype first appearing in China was detected in Korea in 2004–2005 but has not been detected since 2014. A recent study suggests that BoRVA circulates widely among dairy calves in China, and the dominant genotype is G6P[1] [26]; however, the most prevalent genotype in Korea is G6P[5], and the P[1] genotype has not yet emerged.

A recent study from Japan investigated the G and P genotypic profiles of BoRVAs on a farm from 2018 to 2020 [27]. Molecular analysis of positive RVAs (*n* = 122) using semi-nested multiplex RT-PCR, followed by sequencing of selected samples, identified G6, G8, G10, P[1], P[5], and P[11] [27]. The most common combined G and P genotypes were G10P[11] (41.8%) (G6P[1], G6P[5], G6P[11], G8P[1], and G8P[11]), albeit at much lower rates [27]. A previous study conducted systematic surveillance of bovine RVAs in diarrheic samples collected in 1987–2000 from calves on 29 Japanese dairy and beef farms in 11 prefectures. The results revealed the existence of bovine RVAs with multiple genotypes; namely, G6P[5] (37%), G10P[11] (30%), G6P[1] (11%), G6P[11] (11%), G10P[5] (9%), and G8P[11] (1%) [28]. Data suggest that bovine RVAs of the G6P[1], G6P[5], G6P[11], and G10P[11] genotypes have been maintained in cattle herds for over 30 years, and that recently, bovine RVAs with the G8P[1] genotype have become widespread in cattle herds in Japan [27].

In Korea, 80% of the G8P[7] and 10% of the G6P[7] genotypes were BoRVA types prevalent in 2004–2005 [9], although detection of G5P[1], G5P[5], and G5P 7] was reported sometimes during this period [7,8]. However, among these BoRVA genotypes, only G5P[1] was identified in a recent report [10]; the remaining genotypes have been replaced by genotypes G6P[5], G6P[11], G8P[5], G10P[11], and G5P[1].

In a previous study in which the effect of vaccination status on the BoRVA-positive rate in pre-weaned Korean native calves was examined, 42.1% (48/114) of those born to vaccinated cows and 42.6% (147/345) of those born to non-vaccinated cows were positive for BoRVA, based on the results of real-time RT-PCR screening [10]. Interestingly, our findings are similar (26.1% and 25.0%, respectively) to those of that study, although the BoRVA-positive rate in the present study was lower. Currently, polyvalent BoRVA (G6 and G10 genotype) vaccines are used to vaccinate cattle in Korea. However, the BoRVA-positive rates of calves with diarrhea in BoRVA-vaccinated and non-vaccinated farms are similar. It is unclear whether this is due to problems in the effectiveness of the BoRVA vaccine currently used in Korea or to other problems on farms that require further investigation. In addition, a continuous and precise assessment of the prevalence of various VP7 and VP4 type-specificities is needed to evaluate vaccine effectiveness and understand whether it is necessary to construct polyvalent RVA vaccines for livestock animals.

## 5. Conclusions

Among BoRVAs, G6P[5] emerged in Korean calves from around 2014. The G6P[5] and G6P[11] genotypes are the main causative agent in 80% and 14.3%, respectively, of Korean calf infections. In general, BoRVAs infect calves aged less than one month, particularly calves younger than 10 days of age. In order to reduce the economic loss to farms caused by bovine rotavirus, continuous monitoring of VP7 and VP4 should be performed, and in addition, new preventive vaccines and treatments for epidemic rotavirus should be developed.

## Figures and Tables

**Figure 1 animals-12-03555-f001:**
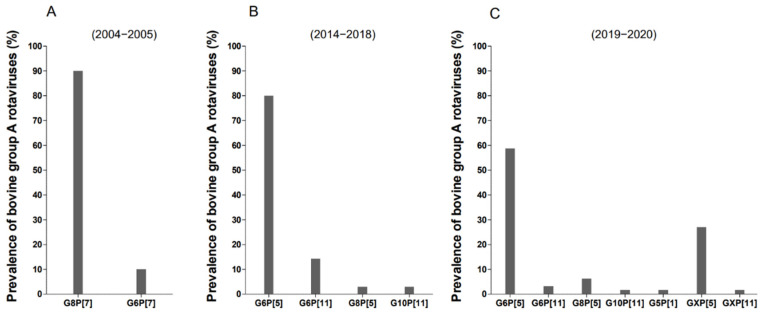
Distribution of the combined G and P BoRVA strains circulating in cows in South Korea. Prevalence of the combined G/P BoRVA strains detected during 2004–2005 (*n* = 30) [9] (**A**), 2014–2018 (*n* = 35) (**B**), and 2019–2020 (*n* = 63) [10] (**C**).

**Figure 2 animals-12-03555-f002:**
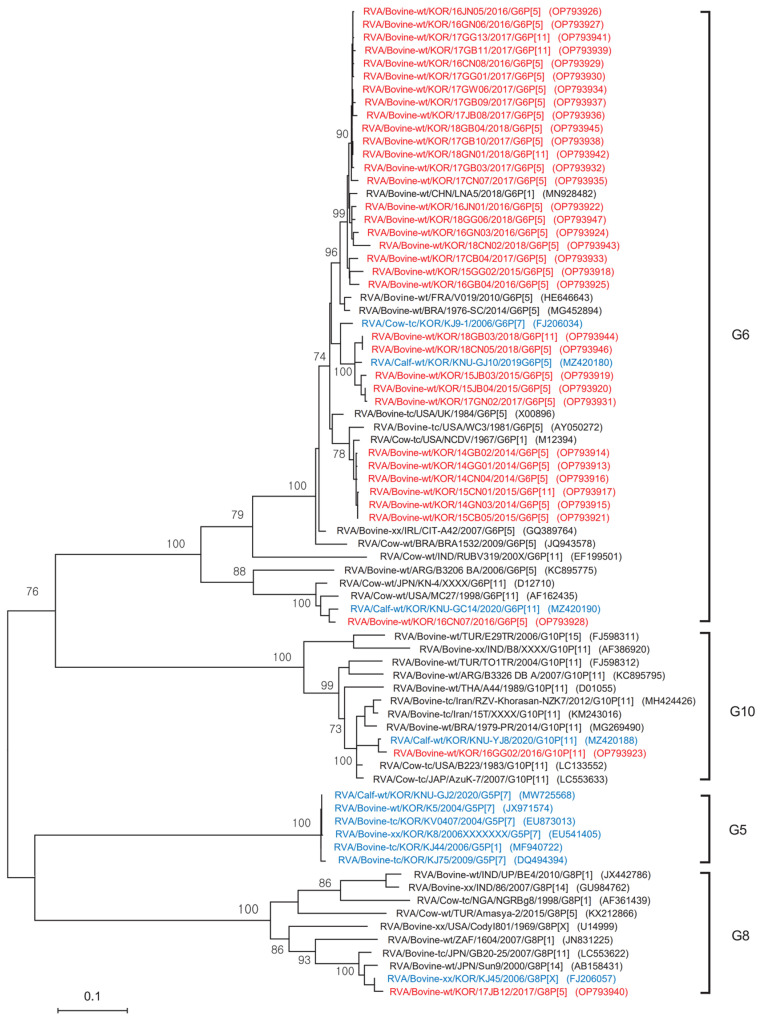
Phylogenetic analysis of a partial nucleotide sequence of the VP7 gene from 35 BoRVA antigen-positive samples and selected reference strains. The phylogenetic tree was constructed using the maximum-likelihood method (based on the Tamura 3-parameter model) in the MEGA X program and tested using 1000 bootstrap values. The 35 Korean BoRVA isolates identified in this study are highlighted in red and Korean BoRVA strains reported by previous studies are highlighted in blue.

**Figure 3 animals-12-03555-f003:**
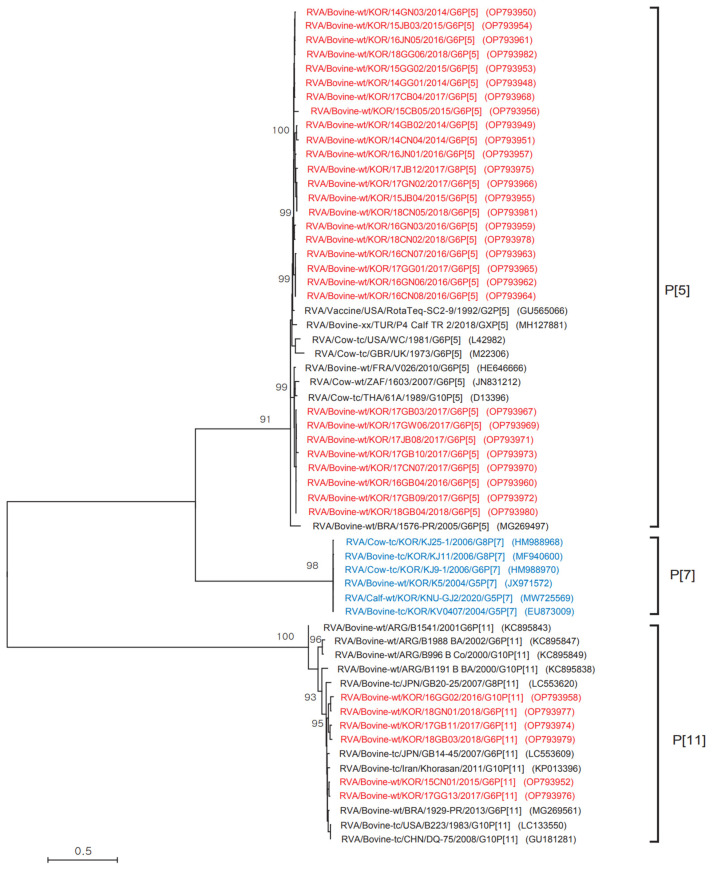
Phylogenetic analysis of a partial nucleotide sequence of the VP4 gene from 35 BoRVA antigen-positive samples and selected reference strains. The phylogenetic tree was constructed using the HKY (Hasegawa–Kishino–Yano) model in the MEGA X program, with 1000 bootstrap values. The 35 Korean BoRVA isolates identified in this study are highlighted in red and Korean BoRVA strains reported by previous studies are highlighted in blue.

**Table 1 animals-12-03555-t001:** Genotyping of 35 BoRVA-positive samples from calf fecal samples collected in South Korea between 2014 and 2018.

Genotype	Number of Bovine Rotavirus-Positive Samples					Total 25.4% (35/138)
	2014 (4/16) *	2015 (5/19)	2016 (8/41)	2017 (12/39)	2018 (6/23)	
G6P[5]	4	4	7	9	4	28
G6P[11]		1		2	2	5
G8P[5]				1		1
G10P[11]			1			1

* Number of BoRVA-positive/number of test samples.

**Table 2 animals-12-03555-t002:** Age, geographical region, and fecal condition of the 35 BoRVA-positive Korean calves sampled between 2014 and 2018.

Strain	Year	VP7 (G Type)	VP4 (P Type)	Province	Fecal Condition	Age (Days)
14GG01	2014	6	5	GG *	Diarrhea	20
14GB02	2014	6	5	GB	Diarrhea	30
14GN03	2014	6	5	GN	Diarrhea	15
14CN04	2014	6	5	CN	Diarrhea	40
15CN01	2015	6	11	CN	Diarrhea	30
15GG02	2015	6	5	GG	Diarrhea	10
15JB03	2015	6	5	JB	Diarrhea	20
15JB04	2015	6	5	JB	Diarrhea	15
15CB05	2015	6	5	CB	Diarrhea	20
16JN01	2016	6	5	JN	Diarrhea	30
16GG02	2016	10	11	GG	Diarrhea	10
16GN03	2016	6	5	GN	Diarrhea	20
16GB04	2016	6	5	GB	Diarrhea	15
16JN05	2016	6	5	JN	Diarrhea	30
16GN06	2016	6	5	GN	Diarrhea	20
16CN07	2016	6	5	CN	Diarrhea	15
16CN08	2016	6	5	CN	Diarrhea	20
17GG01	2017	6	5	GG	Diarrhea	40
17GN02	2017	6	5	GN	Diarrhea	10
17GB03	2017	6	5	GB	Diarrhea	20
17CB04	2017	6	5	CB	Diarrhea	15
17GW06	2017	6	5	GW	Diarrhea	15
17CN07	2017	6	5	CN	Diarrhea	20
17JB08	2017	6	5	JB	Diarrhea	10
17GB09	2017	6	5	GB	Diarrhea	15
17GB10	2017	6	5	GB	Diarrhea	30
17GB11	2017	6	11	GB	Diarrhea	20
17JB12	2017	8	5	JB	Diarrhea	10
17GG13	2017	6	11	GG	Diarrhea	15
18GN01	2018	6	11	GN	Diarrhea	30
18CN02	2018	6	5	CN	Diarrhea	15
18GB03	2018	6	11	GB	Diarrhea	20
18GB04	2018	6	5	GB	Diarrhea	15
18CN05	2018	6	5	CN	Diarrhea	10
18GG06	2018	6	5	GG	Diarrhea	10

* GG: Gyeonggi; GW: Gangwon; CB: Chungbuk; CN: Chungnam; GB: Gyeongbuk; GN: Gyeongnam; JB: Jeonbuk; JN: Jeonnam.

**Table 3 animals-12-03555-t003:** Prevalence of combined genotypes (according to age and region) in 35 BoRVA-positive Korean calves sampled between 2014 and 2018.

Age (Days)	Region	Number of RVA-Positive/Number of Samples	Combined Genotype			
			G6P[5]	G6P[11]	G8P[5]	G10P[11]
1–10		7/20	5		1	1
11–20		20/83	17	3		
21–30		6/22	4	2		
31–50		2/13	2			
	GG *	6/22	4	1		1
	GW	1/8	1			
	CN	7/23	6	1		
	CB	2/10	2			
	GB	8/29	6	2		
	GN	5/20	4	1		
	JB	4/15	3		1	
	JN	2/11	2			

* GG: Gyeonggi; GW: Gangwon; CB: Chungbuk; CN: Chungnam; GB: Gyeongbuk; GN: Gyeongnam; JB: Jeonbuk; JN: Jeonnam.

## Data Availability

The nucleotide sequences of the 2014–2018 viruses obtained in this study were submitted to the GenBank database under accession numbers OP793913-OP793947 for the VP7 gene and OP793948-OP793982 for the VP4 gene.

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
