# Peer review of "Genetic Diversity of Bovine Group A Rotavirus Strains Circulating in Korean Calves during 2014 and 2018"

_animals, 2022, doi:10.3390/ani12243555_

Round 1

Reviewer 1 Report

This manuscript describes the prevalence of BoRVA in the Republic of KOrea.

Unfortunately, this manuscript is not novelty due to the small sample numbers.

THis does not give any information. 

Author Response

Reviewer 1

This manuscript describes the prevalence of BoRVA in the Republic of Korea.

Unfortunately, this manuscript is not novelty due to the small sample numbers.

This does not give any information. 

Answer: Although the number of samples may appear small to the reviewer, we still think that the results are valuable and provide useful information to readers, because samples were collected over several years (2014–2018) and from eight different regions nationwide.

Reviewer 2 Report

Review of Park et al Genetic diversity of bovine group A rotavirus strains circulating in Korean calves during 2014 and 2018.

Park et al have carried out an epidemiological surveillance of different regions of different bovine rotavirus strains circulating among Korean calves during the 2014-2018 seasons. They clearly identify the importance of carrying out such surveillance due to the high economic costs incurred to the beef and dairy industry. Upon review of this article, I provide here my suggestions and my recommendation that will aid towards publication.

Minor Questions/Comments include:

Line 67: Apart from diarrhea, were there any other signs of morbidity observed among the sampled calves? Did any of the sampled calved succumb to RVA disease?

Line 114: replace porcine with bovine

Line 164: an 887 bp fragment instead of an 887 fragment

Line 267: The addition of a concluding statement with an emphasis towards the development of novel prophylaxis and therapeutics would aid further research initiatives in the field of animal rotaviruses.

Author Response

Reviewer 2

Review of Park et al Genetic diversity of bovine group A rotavirus strains circulating in Korean calves during 2014 and 2018.

Park et al have carried out an epidemiological surveillance of different regions of different bovine rotavirus strains circulating among Korean calves during the 2014-2018 seasons. They clearly identify the importance of carrying out such surveillance due to the high economic costs incurred to the beef and dairy industry. Upon review of this article, I provide here my suggestions and my recommendation that will aid towards publication.

Minor Questions/Comments include:

Comment 1: Line 67: Apart from diarrhea, were there any other signs of morbidity observed among the sampled calves? Did any of the sampled calved succumb to RVA disease?

Answer: We have added that “all cattle from which samples were taken showed signs of diarrhea and dehydration, along with anorexia and inactivity. Respiratory symptoms were also observed in some calves.” (lines 73-75).

Comment 2: Line 114: replace porcine with bovine

Answer: We have changed “porcine” to “bovine” (line 113).

Comment 3: Line 164: an 887 bp fragment instead of an 887 fragment

Answer: We changed “887 fragment” to “887 bp fragment” (lines 162-163)

Comment 4: Line 267: The addition of a concluding statement with an emphasis towards the development of novel prophylaxis and therapeutics would aid further research initiatives in the field of animal rotaviruses.

Answer: We added a sentence (lines 267-270).

Reviewer 3 Report

animals-2058729

General comments:

In this manuscript entitled “Genetic diversity of bovine group A rotavirus strains circulating 2 in Korean calves during 2014 and 2018”, the authors investigated annual changes in BoRVA strains by examining the VP4 and VP7 genes of rotaviruses in Korean calves. This study showed that the G6P genotype is the main causative agent of diarrhea in Korean calves. In addition, it is predicted that the G6P will continue to act as a major cause of diarrhea in Korean calves. While the findings presented in this study should be of interest to Animals readership, it requires some revision before ready for publication.

General comments:

The authors are using the words “prevalence BoRVA” and “BoRVA-positive rate” in the manuscript. The terms should be unified in the same manner. BoRVA-positive rate is more correct in this manuscript so that it should be changed to it. 

Specific comments:

Line 243: I could not find the reference number [28] in the References.

Line 252-262: Authors are discussing the BoRVA vaccine used in Korea. What types of vaccine are used in Korea? Should explain the vaccine more detailed.

Line 255: “positive for BoRVA” Do you mean any diagnostic test for BoRVA showed positive? What kind of the test was conducted?

Line 257-258: “These results may mean that the current BoRVA vaccine is no longer effective in the field.” I could not understand this logic. Why can you say that? You should explain more logically about this sentence.

Line 341: Is the reference number “28” missing?

Table 1: This should not be separated with two pages, should be placed in one page.  

Author Response

Reviewer 3

General comments:

In this manuscript entitled “Genetic diversity of bovine group A rotavirus strains circulating 2 in Korean calves during 2014 and 2018”, the authors investigated annual changes in BoRVA strains by examining the VP4 and VP7 genes of rotaviruses in Korean calves. This study showed that the G6P genotype is the main causative agent of diarrhea in Korean calves. In addition, it is predicted that the G6P will continue to act as a major cause of diarrhea in Korean calves. While the findings presented in this study should be of interest to Animals readership, it requires some revision before ready for publication.

General comments:

The authors are using the words “prevalence BoRVA” and “BoRVA-positive rate” in the manuscript. The terms should be unified in the same manner. BoRVA-positive rate is more correct in this manuscript so that it should be changed to it. 

Answer: We changed “prevalence BoRVA” to “BoRVA-positive rate” throughout the revised manuscript.

Specific comments:

Comment 1: Line 243: I could not find the reference number [28] in the References.

Answer: We added the reference to the list of references (line 241).

Comment 2: Line 252-262: Authors are discussing the BoRVA vaccine used in Korea. What types of vaccine are used in Korea? Should explain the vaccine more detailed.

Answer: We revised the sentences (lines 255-262).

Comment 3: Line 255: “positive for BoRVA” Do you mean any diagnostic test for BoRVA showed positive? What kind of the test was conducted?

Answer: As in the previous study (reference number 10), BoRVA positive was confirmed by real-time RT-PCR screening (line 253).

Comment 4: Line 257-258: “These results may mean that the current BoRVA vaccine is no longer effective in the field.” I could not understand this logic. Why can you say that? You should explain more logically about this sentence.

Answer: We revised the sentences (lines 255-262).

Comment 5: Line 341: Is the reference number “28” missing?

Answer: We added the reference to the reference list (lines 344-345).

Comment 6: Table 1: This should not be separated with two pages, should be placed in one page.  

Answer: We have adjusted table 1 so that it appears on only one page.

Round 2

Reviewer 1 Report

None